# The triple burden of malnutrition among adolescents in Dar es Salaam, Tanzania: The role of gender, household environment, and food insecurity

**Christina Kimaryo** [1*], **Josephine Efraim**[1], **Nancy Njenge**[1], **Bruno Sunguya**[2], **Lyn Haskins**[3], **Anne Hatløy** [1,4], **Christiane Horwood**[3]

**1** Centre for International Health, University of Bergen, Bergen, Norway, **2** Department of Community Health, School of Public Health and Social Sciences, Muhimbili University of Health and Allied Sciences, Dar es Salaam, Tanzania, **3** Centre for Rural Health, University of KwaZulu-Natal, Durban, South Africa, **4** Fafo Institute for Labour and Social Science, Oslo, Norway

* christinekimaryo@gmail.com

## Abstract

### Background

Adolescence is a critical growth period with increased nutritional needs that can put adolescents at risk for the triple burden of malnutrition, which can impair brain development, reduce functional capacity, and heighten the risk of non-communicable diseases. Despite this, there is limited data on factors associated with adolescent malnutrition in Dar es Salaam. This study aimed to determine the prevalence and associated factors for anaemia, stunting, underweight and overweight/obesity among adolescents in Dar es Salaam.

### Methods

A cross-sectional study was conducted among 507 adolescents aged 12–19 in Dar es Salaam, using multistage cluster sampling. Data was collected using a pre-validated structured questionnaire through an Open Data Kit, while anthropometric measurements were measured using a digital weighing scale and stadiometer. Hemocue machines were utilised to measure anaemia through capillary blood samples. Data analysis, including descriptive statistics and modified Poisson regression, was performed using Stata v18 to identify factors associated with malnutrition.

### Results

Overweight affected 16%, anaemia 39%, stunting 23% and thinness 12% of all adolescents. Food insecurity affected 69% of the households and was associated with a lower overweight risk (Crude Prevalence Ratio (CPR) 0.51, 95% CI: 0.34–0.75, p = 0.001) and a higher risk for thinness (CPR 2.10, 95% CI: 1.13, 3.91, p = 0.019).

**Data availability statement:** All data files are available from the Zenodo database (Accessed at https://doi.org/10.5281/zenodo.16929962).

**Funding:** This study was conducted as part of EPRENUT, a NORPART project funded by HK Dir Norway (NORPART-2021/10434). The funders had no role in study design, data collection and analysis, decision to publish, or preparation of the manuscript.

**Competing interests:** The authors have declared that no competing interests exist.

After controlling for other explanatory variables, adolescents from households with unimproved toilet facilities had a higher risk of anaemia (APR 1.51, 95% CI: 1.16–1.95, p = 0.002). Girls had a lower risk of stunting (APR 0.63, 95%, CI: 0.45–0.88, p = 0.006) and thinness (APR 0.29, 95%, CI: 0.16–0.54, p < 0.001) but a higher risk for overweight (APR 2.62, 95%, CI: 1.56–4.43, p < 0.001) compared to boys.

## Conclusion

Adolescents in Dar es Salaam experience high rates of malnutrition. Food insecurity, poor sanitation, and gender-specific factors contribute significantly to these outcomes. Integrated interventions are essential to address these interlinked determinants of malnutrition in this vulnerable group.

## Introduction

Adolescence is a critical stage in the human life cycle, marking the transition from childhood to adulthood. It is characterised by rapid, simultaneous changes, including physical growth, such as increases in height and weight, as well as significant cognitive, emotional, and psychosocial development, encompassing cognitive advancement, emotional regulation, and social maturation [1]. This life stage provides a good window of opportunity for addressing global and intergenerational health problems [2,3]. In this study, adolescence is defined as individuals aged 12–19 years, following the World Health Organisation's definition of 10–19 years, but restricted to those aged 12 years and above to ensure sufficient cognitive maturity and decision-making capacity for informed participation [4].

The triple burden of malnutrition, defined as the coexistence of overnutrition, undernutrition and micronutrient deficiencies within the same community, is an increasing trend among adolescents in low and middle-income countries (LMICs) [5]. In this era of epidemiological, demographic and nutritional transitions, micronutrient deficiencies, stunting, and underweight remain prevalent in LMICs, whilst overweight and obesity are increasing [6]. This is exacerbated by rapid urbanisation, availability of cheap processed food and reduced physical exercise [7].

Multiple factors contribute to malnutrition, including cultural and gender norms, which often discriminate against adolescent girls, like discouraging girls from engaging in physical activities and societal norms that encourage early marriages and pregnancies [8]. Structural policy shortcomings, such as the failure to incorporate food security, access to healthy foods, and safe public spaces for physical activity in urban settings, contribute to the growing burden of malnutrition among populations in urban and socioeconomically deprived areas [9]. Other factors associated with malnutrition include parental education, place of residence (rural/urban), household income, availability of toilet facilities, household size greater than five and household food security [10–12].

Adolescent malnutrition can be detrimental to adolescents' current and future health, hampering brain growth and functional capacity and making them vulnerable

to non-communicable diseases (NCDs) in later life [1,8], There are other severe long-term effects of adolescent malnutrition, for example, stunting is associated with impaired cognitive development and decreased economic productivity [13], obesity is associated with type 2 diabetes and hypertension [14], and anaemia with reduced cognitive function [8]. Anaemia in female adolescents is particularly important because of the association with poor pregnancy outcomes in settings with high rates of teenage pregnancy [15].

Globally, the prevalence of adolescent anaemia is estimated to be 15%, and 8% of females and 12% of males are underweight [16,17]. There are large disparities in the prevalence of adolescent malnutrition between high and low-income countries. Like other low-income countries, Tanzania is facing this complex triple burden of malnutrition. A study conducted in Tanzania mainland in 2023 found a 34% prevalence of anaemia, 32% prevalence of stunting and 4% prevalence of obesity among adolescents [18]. The high prevalence of malnutrition among school-going adolescents was associated with modifiable factors including poor diet quality, food insecurity and unavailability of school handwashing stations [2].

Adolescents are often a forgotten group in policy and priority settings, especially regarding adolescents' nutrition. Additionally, existing studies on adolescent nutrition in the region have largely been conducted among school-going adolescents, limiting the generalizability of their findings to the broader adolescent population. It is important to quantify the prevalence of malnutrition among adolescents, including those who are not attending school, and explore associated factors for policymakers to design appropriate interventions. Therefore, we conducted a household survey aimed at assessing the prevalence and factors associated with the triple burden of malnutrition among adolescents in Dar es Salaam, Tanzania.

## Methodology

### Study design

A cross-sectional analytical household study was conducted, consisting of structured interviews with adolescents and their guardians, as well as measurements of adolescents' height, weight and haemoglobin levels.

### Study setting

This study was conducted in Dar es Salaam City, which is one of 5 councils in the Dar es Salaam region, Tanzania's largest city and a major economic centre. Dar es Salaam City is the most populated council with 1.6 million inhabitants [19]. Adolescents aged 13–19 make up 13% of the population in the Dar es Salaam region [19]. In 2022, the unemployment rate in the Dar es Salaam region was 17%, and most people (57%) in this region were working in the private sector [19]. Among school-going adolescents in Dar es Salaam, there was a high prevalence of anaemia at 58% [2], while 14% of girls were affected by overweight or obesity, and 13% of boys were underweight [3].

### Study population

Eligible participants were adolescents aged 12–19 and their guardians residing in Dar es Salaam City at the time of the survey. Pregnant adolescents and those who were less than four months postpartum were excluded from the study as BMI is not reliable during this period, and anaemia cut-off points are different for pregnant vs non-pregnant women [20].

### Sampling and sample size calculation

A sample size of 540 was calculated in this study, assuming a 14% prevalence of overweight and obesity observed in a previous study from Dar es Salaam [3], the power of 80%, a probability of 95%, a non-response rate of 10% and a design effect of 1.2 to adjust for clustering.

A multi-stage cluster sampling was used based on the National Bureau of Statistics (Tanzania) enumeration areas in Dar es Salaam City. We had a list of all 36 wards in Dar es Salaam city and the total number of households in each ward. Forty-five clusters were selected as the first-stage primary sampling units (PSUs) using a probability proportional to size

(PPS) approach. In the second stage, 12 households were randomly selected within each cluster. From each selected household, one eligible adolescent aged 12–19 years was randomly chosen to participate in the study, resulting in a total sample of 507 adolescents. Households were randomly selected using a random walk method to ensure a realistic representation of all adolescents in the city. The starting point for the random walk was determined by identifying the centre of each cluster's map, where a prominent landmark such as a church, school, or market was identified. Data collectors began their walk from this point, using a pen to indicate the direction to proceed by throwing the pen in the air and following the direction in which it landed. Every third household in the chosen direction was approached for an interview. In cases where a household did not have any eligible participants, the third subsequent household was selected until an adolescent was encountered. For households with adolescents who were absent during data collection, appointments were arranged for interviews when they were available.

## Data collection and tools

Data were collected by trained enumerators. A three-day training was conducted for enumerators covering ethical procedures, questionnaire administration, and accurate anthropometric measurements. The training included practical sessions and role-playing to ensure consistency and standardisation. After training, we pre-tested the questionnaire with our trained research assistants to assess its cultural appropriateness, translation accuracy and duration. The questionnaire was adapted as necessary based on the findings of the pre-test. Details of the questionnaire are provided (S1 File).

Data collection took place in Dar es Salaam City from September 4 to September 17, 2023. Data was collected electronically by 16 trained research assistants using a validated questionnaire through an Open Data Kit (ODK). The structured questionnaire used to gather information on household characteristics and socio-economic status was adapted from the Tanzania Demographic and Health Survey (TDHS) [21]. Household-level sociodemographic information, such as parental education and household size, was provided by the adolescent´s guardians. In contrast, all individual-level data, including sociodemographic details and health outcomes, were collected directly from the adolescents. All interviews were performed in Swahili, and all data collection procedures were conducted according to the study's standard operating procedures (SOP).

Weight and height were measured using a digital weighing scale (SECA) and a wooden stadiometer, respectively. Each participant had their weight and height measured twice to reduce measurement error. Hemoglobin concentration was measured using portable point-of-care devices, the HemoCue Hb 201+ and the HemoCue Hb 801 systems [22]. Since data collection took place outdoors in Dar es Salaam, we took care to conduct measurements in shaded areas to limit device exposure to direct sunlight and extreme heat, in accordance with manufacturer guidelines [22]. No altitude adjustments were applied to the haemoglobin values, as Dar es Salaam is located at sea level below the 1000-meter threshold recommended by the World Health Organization (WHO) for altitude correction [23]. Additionally, each field team was supervised daily by field supervisors to ensure adherence to study protocols and resolve any issues in real-time and ensure data quality. Daily data backups and routine reviews by the study team helped monitor data integrity and consistency across teams.

The choices for independent variables were guided by the UNICEF conceptual framework for factors affecting adolescent nutrition [24]. The validated USAID Household Food Insecurity Access Scale (HFIAS) was used to assess food insecurity. The HFIAS comprises nine questions related to access to food. If any answers indicated that there was a lack of access to food, participants were asked how often this occurred (rarely, sometimes, often). A score was calculated based on these answers and categorised into no food insecurity, mild/moderate food insecurity, or severe food insecurity [25].

Food insecurity categories were re-grouped into two groups (food secure vs. food insecure) due to small sample sizes in some subgroups, which limited statistical power for detecting meaningful differences across all four levels.

Physical activity was captured using questions about the number of days spent on activities like any play, game, sport, walking, exercise at home/work or school, and whether the adolescent performs home chores or not. The questions were

adapted from a validated questionnaire from the International Study of childhood obesity, Lifestyle and Environment (ISCOLE) [26]. Individual health status was assessed through adolescents' self-rating their health as either poor, fair, good or excellent. Both physical activity and individual health status were recorded as obtained from the questionnaire without any changes.

The main outcome variables were anaemia, thinness, stunting and overweight/obesity. Anaemia was defined by WHO guidelines, using haemoglobin cut-offs for age and sex [22]. Overweight/obesity and thinness were defined using WHO BMI-for-age-Z-scores; and stunting using Height for age Z-scores (HAZ<2SD) [27]. Overweight/obesity was defined as BMI≥+1SD, thinness as ≤−2SD.

### Data analysis

Stata version 18 was used for data cleaning and analysis. Categorical variables were summarized using frequencies and percentages, and continuous variables were summarized using the mean and standard deviation. WHO Anthroplus software (R Package) was used to compute BMI-for-age z-scores and height-for-age z-scores [28]. Pearson Chi-square test was used to characterize the burden of overweight, thinness, stunting and anaemia. A modified Poisson regression model was used to test associations between predictor variables and the outcome variable at the bivariate and multivariate levels. All the predictor variables were put in the full model (multivariate analysis) as confounders. **Table 1** shows how all covariates included in the multivariable models were defined and standardised.

The household wealth index was derived through principal component analysis (PCA). Initially, a selection of potential wealth indicators, encompassing housing characteristics and owned household assets and goods, was made. The item-rest correlation for each indicator was then examined, with a minimum threshold of 0.1 considered for inclusion in the PCA. Additionally, Cronbach's alpha was utilized to assess the correlation coefficients of each item, with items having an alpha value of 0.7 or higher considered acceptable [29]. The final set of items used in the analysis to construct wealth indexes comprised access to electricity, ownership of a radio, television, computer, non-mobile telephone, refrigerator, microwave, iron, oven, watch, bicycle, and motorcycle. The results were then categorised into "low" "middle" and "high".

### Ethical consideration

Data was stored anonymously using encrypted codes in a secure server for research data at the University of Bergen (SAFE), where only authorised personnel are allowed to access it. For transparency, we have shared the anonymised data on a public repository at https://doi.org/10.5281/zenodo.16929962. Ethical approval was obtained from the National Institute for Medical Research (NIMR) in Tanzania (Ref: NIMR/HQ/R.8a/Vol.IX/4415), Muhimbili University of Health and Allied Sciences (MUHAS) (Ref. No.DA.282/298/01.C/1790) and the Regional Ethical Committee (REC) in Norway (Ref: 614998). Additionally, we obtained permits from local authorities, including the Minister of State, President's Office (Regional Administration and Local Government) Office and the Ilala district medical officer. Informed written consent from parents and adolescents above 18 years and written assent from adolescents below 18 years were obtained after being informed of the potential risks associated with the study. Privacy, anonymity, and confidentiality were ensured throughout the process of the study.

### Inclusivity in global research

Additional information regarding the ethical, cultural, and scientific considerations specific to inclusivity in global research is included in the Supporting Information (S1 Checklist).

## Results

### Sociodemographic characteristics

The survey had a response rate of 96.8%. Adolescents in 507 out of 540 households agreed to participate in the survey, while 33 adolescents did not give their consent/assent. **Table 2** presents the characteristics of the study participants.

**Table 1.** Independent variables and their descriptions/categorization.

| Characteristics | Category/Measure | Description/Definition |
|---|---|---|
| Adolescent Age | 12-14 years "0"<br>15-19 years "1" | Age in completed years at time of survey |
| Adolescent Gender | Male "1", Female "2" | Biological sex of the participant |
| Ever attended school | Yes "1"/No "0" | Whether an adolescent ever attended school |
| Currently attending school | Yes"1"/No "0" | Whether the adolescent was enrolled at the time of the survey |
| Adolescent school level | Primary, "0" Secondary/higher "1" | Highest school level attended or attending |
| Adolescents working for a living | Yes "1"/No "0" | Self-reported employment status |
| General health status | Poor/ fair "1" Good/Excellent "2", No answer "3" | Self-rated health status |
| Chronic condition or disability | Yes "1"/No "0" | Self-reported presence of chronic illness/disability |
| Currently attending clinic for chronic condition | Yes "1"/No "0" | Currently attending a health facility for a chronic condition. |
| Diagnosed mental illness | Yes "1"/No "0" | Self-reported diagnosis by a health professional |
| Played sports in the past 7 days | Yes "1"/No "0" | Any physical sports activity last week |
| Days spent on sports | None "0", 1–2 days "1", 3–5 days "2", 6–7 days "3" | Frequency of sport participation in the past 7 days |
| Time taken to walk to school | None "0", 5–30 minutes "1", >30 minutes "2" | Self-reported time walking to school |
| Perform house chores | Yes "1"/No "0" | Self-reported participation in chores like gardening |
| Household wealth index | Low "1", Middle "2", High "3" | Generated via PCA and categorised into tertiles |
| Household head's education level | primary/no formal education "1", Secondary/higher "2" | The highest education attained by the household head |
| The household's main source of water | Improved "1, Unimproved"2" | Based on the WHO/UNICEF water source classification |
| Household toilet facility | Improved "1", Unimproved "2" | Based on the WHO/UNICEF sanitation classification |
| Household size | <6 members "0"<br>≥6 members "1" | Number of people in the household |
| HFIAS food insecurity categories | Food secure "0",<br>Food insecure (mild, moderate and severe) "1" | Measured using the Household Food Insecurity Access Scale |

Notably, nearly 70% of households were classified as severely food insecure. Additionally, 83 (16%) households had unimproved toilet facilities.

## Prevalence of malnutrition

The overall prevalence of overweight among all participants was 16% (82/507). This was higher among girls (64/274; 23%) compared to boys (18/233;8%, p<0.001). However, boys had a higher prevalence of thinness compared to girls (50/233; 21% vs 15/274; 5%, p<0.001).

Anaemia affected 200/507 (39%) of all adolescents. Girls were more affected by anaemia than boys (123/274; 45% vs 77/233; 33%, p=0.007). Stunting affected about a quarter of all adolescents (120/507, 24%) (**Table 3**). Co-existing forms of malnutrition were measured in the same individual at the same time.

## Bivariate analysis of factors associated with anaemia, overweight, stunting and thinness

Higher risks of anaemia were observed among female adolescents (CPR 1.36, 95% CI: 1.08–1.70, p=0.008) and those in households with unimproved toilet facilities (CPR 1.61, 95% CI: 1.29–2.02, p<0.001). Female adolescents had a significantly higher risk for overweight than males (CPR 3.02, 95% CI: 1.85–4.95, p<0.001), as did adolescents from middle- and high-income households compared to low-income (Middle: CPR=1.89; p=0.017; High: CPR=1.94; p=0.020).

**Table 2. Respondents' characteristics.**

| Characteristic | Frequency N = 507 | Percent |
|---|---|---|
| **ADOLESCENTS´ CHARACTERISTICS** | | |
| **Adolescent Age** | | |
| Mean (SD) 15.2 (±2.0) | | |
| 12-14 | 241 | 47.5 |
| 15-19 | 266 | 52.5 |
| **Adolescent Gender** | | |
| Male | 233 | 46.0 |
| Female | 274 | 54.0 |
| **Ever attended school** | 502 | 99.0 |
| **Currently attending school** | 426 | 84.0 |
| **Adolescent school level (n = 426)** | | |
| Primary education | 165 | 38.7 |
| Secondary/higher | 261 | 61.3 |
| **Adolescents working for a living** | 42 | 8.3 |
| **In general, how do you consider your health?** | | |
| Excellent | 53 | 10.5 |
| Good | 374 | 73.8 |
| Fair | 43 | 8.5 |
| Poor | 31 | 6.1 |
| No answer | 6 | 1.2 |
| **Have any chronic condition or disability** | 70 | 13.8 |
| **Currently, attending a clinic for a chronic health condition (n = 70)** | 14 | 20.0 |
| **Has ever been diagnosed with a mental illness** | 4 | 0.8 |
| **Physical activity** **Has played any kind of sport in the past seven days (n = 506)** | 365 | 72.1 |
| **How many days have you spent on sports?** | | |
| None | 145 | 28.7 |
| 1–2 days | 117 | 23.1 |
| 3–5 days | 115 | 22.7 |
| 6–7 days | 129 | 25.5 |
| **Time taken to walk to school (n = 501) *** | | |
| None | 124 | 24.8 |
| 5–30 minutes | 291 | 58.1 |
| Above 30 minutes | 86 | 17.1 |
| **Perform house chores, e.g., Gardening** | 458 | 90.3 |
| **HOUSEHOLD CHARACTERISTICS** | | |
| **Household wealth index** | | |
| Low | 177 | 34.9 |
| Middle | 198 | 39.1 |
| High | 132 | 26.0 |
| **Household head's education level** | | |
| No formal education | 18 | 3.6 |
| Primary education | 301 | 60.0 |
| Secondary/higher education | 188 | 37.1 |

*(Continued)*

**Table 2.** (Continued)

| Characteristic | Frequency N = 507 | Percent |
|---|---|---|
| **The household's main source of water** | | |
| Improved water sources | 488 | 96.3 |
| Unimproved water sources | 19 | 3.7 |
| **Household toilet facility** | | |
| Improved toilet facilities | 424 | 83.6 |
| Unimproved toilet facilities | 83 | 16.4 |
| **Household size** | | |
| Mean (SD) 5.8 (±2.4) | | |
| <6 | 254 | 50.1 |
| >=6 | 253 | 49.9 |
| **HFIAS food insecurity categories** | | |
| Food secure | 157 | 30.9 |
| Any food insecurity | 350 | 69.0 |
| Mildly food insecure | 39 | 7.7 |
| Moderately food insecure | 123 | 24.3 |
| Severely food insecure | 188 | 37.1 |

*Adolescents' school level was asked among those currently attending school (n = 426).

**Table 3. Prevalences of overweight, thinness, stunting, anaemia and individual DBM.**

| Nutritional status | Male (N = 233) | | Female (N = 274) | | Total (N = 507) N (N = 507(n = 507) | | P-values |
|---|---|---|---|---|---|---|---|
| | n | % | n | % | n | % | |
| **No malnutrition** | 89 | 38.2 | 87 | 31.8 | 176 | 34.7 | 0.129 |
| **Anaemia** | 77 | 33.0 | 123 | 44.9 | 200 | 39.4 | 0.007 |
| **Categories of anaemia** | | | | | | | |
| No anaemia | 156 | 67.0 | 151 | 55.1 | 307 | 60.6 | <0.001 |
| Mild anaemia | 65 | 27.9 | 71 | 25.9 | 136 | 26.8 | |
| Moderate anaemia | 12 | 5.2 | 45 | 16.4 | 57 | 11.2 | |
| Severe anaemia | 0 | 0.0 | 7 | 2.6 | 7 | 1.4 | |
| **Overweight** | 18 | 7.7 | 64 | 23.4 | 82 | 16.2 | <0.001 |
| **Thinness** | 50 | 21.5 | 15 | 5.5 | 65 | 12.8 | <0.001 |
| **Stunting** | 68 | 29.2 | 52 | 18.9 | 120 | 23.7 | 0.007 |
| **Co-existing anaemia and overweight** | 5 | 2.1 | 28 | 10.2 | 33 | 6.5 | <0.001 |
| **Co-existing stunting and overweight** | 6 | 2.6 | 13 | 4.7 | 19 | 3.8 | 0.200 |
| **Co-existing stunting and anaemia** | 30 | 12.9 | 24 | 8.8 | 54 | 10.7 | 0.134 |
| **Co-existing thinness and anaemia** | 20 | 8.6 | 9 | 3.3 | 29 | 5.7 | 0.010 |
| **Co-existing overweight, anaemia and stunting** | 2 | 0.9 | 7 | 2.6 | 9 | 1.8 | 0.149 |
| **Co-existing thinness, anaemia and stunting** | 8 | 3.4 | 2 | 0.7 | 10 | 1.9 | 0.029 |

*P-values based on Chi-square tests. Significant p-values (<0.05) suggest a significant difference in malnutrition prevalence between male and female adolescents.

*Cut-offs for anaemia are Haemoglobins in g/dl, for overweight& thinness is BMI-for-age-Z-scores and stunting is Height-for- age-Z-scores.

*Anaemia is a combination of mild, moderate and severe anaemia.

Engaging in sports was associated with a reduced overweight risk (CPR = 0.49, CI: 0.33–0.73, p < 0.001), as was longer walking times to school (CPR = 0.35, CI: 0.17–0.72, p = 0.004). Food insecurity (at all levels) was associated with lower risk for overweight (CPR 0.51, 95% CI: 0.34–0.75, p = 0.001).

Larger household size (> 6 people) was associated with the increased risk of stunting (CPR 1.07, 95% CI: 1.02–1.13, p = 0.009), while females (CPR 0.65, 95% CI: 0.47–0.89, p = 0.008) and adolescents attending school (CPR 0.69, 95% CI: 0.48–0.99, p = 0.044) had reduced risk of stunting. Higher risks of thinness were observed among adolescents from food-insecure households (CPR 2.10, 95% CI: 1.13–3.91, p = 0.019) and those engaging in sports (CPR = 2.75, CI: 1.35–5.62 p = 0.005) and walking more than 30 minutes to school (CPR 4.61, 95% CI: 1.75–12.13, p = 0.002). Female adolescents had lower risks of thinness (CPR 0.26, 95% CI: 0.15–0.44, p = 0.001), as did those from high-income households (CPR 0.27, 95% CI: 0.11–0.63, p = 0.002), as shown in **Table 4**.

### Multivariate analysis of factors associated with anaemia, overweight, stunting and thinness

After controlling for other variables, we found that adolescents from households with unimproved toilet facilities had a higher risk of anaemia (APR 1.51, 95% CI: 1.16–1.95, p = 0.002) compared to those from improved facilities. Regarding overweight, females showed an increased risk of overweight compared to males (APR 2.62, 95% CI: 1.56–4.43, < 0.001) while adolescents who came from Food-insecure households (any level) had a lower risk (APR 0.59, 95% CI: 0.38–0.92, p = 0.020). Residing in a larger household (>6 people) was associated with an increased risk of stunting (APR 1.06, 95% CI: 1.01–1.11, p = 0.010) while females had a lower risk of stunting compared to males (APR 0.63, 95% CI: 0.45–0.88, p = 0.006). For thinness, Females and those coming from high-income households had a significantly lower risk than males and those from low-income households (APR 0.29, 95% CI: 0.16–0.54, p < 0.001and APR 0.39, 95% CI: 0.16–0.96, p = 0.040), respectively **(Table 5)**.

## Discussion

Adolescent malnutrition remains a critical public health issue in Dar es Salaam, with this study highlighting various individual and household-level factors associated with malnutrition. Our study found that overweight affected approximately 1 in 6 adolescents, with adolescent girls over three times more affected than their male counterparts. This gender disparity reflects broader structural and sociocultural norms in urban LMICs, where gendered expectations and limited access to safe public spaces often restrict girls' participation in physical activity [30]. These disparities highlight the need for gender-specific interventions that go beyond individual behaviours to address the underlying cultural and social norms influencing dietary intake, physical activity, and access to health resources. For instance, promoting girls' participation in sports and physical education programs could help mitigate some of the risks. Moreover, interventions should be inclusive of both in-school and out-of-school adolescents, ensuring that no group is left behind in efforts to reduce gender-based nutritional inequities.

Nearly 7 out of 10 households in the study experienced food insecurity. This is a major public health concern, as food insecurity is associated with both undernutrition and overweight, reflecting the double burden of malnutrition commonly observed in rapidly urbanising LMICs, where reduced physical activities, limited dietary diversity, rapid changes in the food system, particularly the availability of cheap ultra-processed food and poverty interact to produce both under- and overnutrition [7,31].

In our study, food insecurity was associated with both overweight and thinness in bivariate analysis. Adolescents from food-insecure households were more likely to be thin and less likely to be overweight, a pattern that has been linked in previous studies to reduced overall food intake [32]. Furthermore, we found that adolescents from households with a higher wealth index had a reduced risk of thinness. The observed association between food insecurity, household wealth index, and thinness highlights the complex ways in which poverty shapes adolescent nutritional outcomes in LMICs, underscoring the need for integrated strategies that address both undernutrition and overnutrition within the same population.

**Table 4. Bivariate analysis of factors associated with anaemia, overweight, stunting, and thinness.**

| Variables | Anaemia | | Overweight | | Stunting | | Thinness | |
|---|---|---|---|---|---|---|---|---|
| | CPR (95 CI) | p-value | CPR (95 CI) | p-value | CPR (95 CI) | p-value | CPR (95 CI) | p-value |
| **Adolescent age** | 1.01 (0.96,1.07) | 0.616 | 1.14 (1.02,1.26) | **0.017** | 0.99 (0.92,1.07) | 0.803 | 0.97 (0.87,1.08) | 0.622 |
| **Household size** | 0.99 (0.95,1.04) | 0.822 | 0.95 (0.87,1.05) | 0.298 | 1.07 (1.02,1.13) | **0.009** | 1.04 (0.96,1.14) | 0.339 |
| **Adolescent Gender** | | | | | | | | |
| Male | 1 | | 1 | | 1 | | 1 | |
| Female | 1.36 (1.08,1.70) | **0.008** | 3.02 (1.85,4.95) | **0.001** | 0.65 (0.47,0.89) | **0.008** | 0.26 (0.15,0.44) | **<0.001** |
| **Currently attending school** | | | | | | | | |
| No | 1 | | 1 | | 1 | | 1 | |
| Yes | 1 (0.74,1.34) | 0.991 | 0.68 (0.42,1.08) | 0.1 | 0.69 (0.48,0.99) | **0.044** | 1.87 (0.83,4.19) | 0.128 |
| **Adolescent school level** | | | | | | | | |
| Primary education | 1 | | 1 | | 1 | | 1 | |
| Secondary/higher | 1.2 (0.93,1.54) | 0.157 | 1.21 (0.75,1.95) | 0.441 | 0.72 (0.50,1.03) | 0.068 | 0.7 (0.44,1.12) | 0.139 |
| **Household Wealth Index** | | | | | | | | |
| Low | 1 | | 1 | | 1 | | 1 | |
| Middle | 1.07 (0.83,1.38) | 0.605 | 1.89 (1.12,3.19) | **0.017** | 0.73 (0.51,1.05) | 0.091 | 0.86 (0.54,1.38) | 0.542 |
| High | 1.12 (0.85,1.48) | 0.434 | 1.94 (1.11,3.38) | **0.020** | 0.78 (0.52,1.16) | 0.216 | 0.27 (0.11,0.63) | **0.002** |
| **Household head education** | | | | | | | | |
| Primary/no formal | 1 | | 1 | | 1 | | 1 | |
| Secondary/higher | 0.84 (0.66,1.06) | 0.132 | 1.7 (1.14,2.52) | **0.009** | 0.82 (0.58,1.14) | 0.24 | 0.55 (0.32,0.95) | **0.031** |
| **Household toilet facility** | | | | | | | | |
| Improved | 1 | | 1 | | 1 | | 1 | |
| Unimproved | 1.61 (1.29,2.02) | **<0.001** | 0.96 (0.56,1.66) | 0.89 | 1.08 (0.72,1.63) | 0.7 | 1.28 (0.73,2.24) | 0.393 |
| **Played any sport in the past 7 days** | | | | | | | | |
| No | 1 | | 1 | | 1 | | 1 | |
| Yes | 0.7 (0.57,0.87) | **0.001** | 0.49 (0.33,0.73) | **<0.001** | 1.27 (0.87,1.85) | 0.214 | 2.75 (1.35,5.62) | **0.005** |
| **Days spent on physical activity** | | | | | | | | |
| None | 1 | | 1 | | 1 | | 1 | |
| 1-2 | 0.81 (0.61,1.07) | 0.132 | 0.67 (0.41,1.09) | 0.107 | 1.45 (0.94,2.24) | 0.09 | 2.62 (1.23,5.57) | **0.013** |
| 3-5 | 0.77 (0.58,1.02) | 0.072 | 0.41 (0.22,0.75) | **0.004** | 1.04 (0.64,1.69) | 0.863 | 2.66 (1.25,5.66) | **0.011** |
| 6-7 | 0.58 (0.42,0.79) | **0.001** | 0.39 (0.22,0.71) | **0.002** | 1.28 (0.82,1.98) | 0.272 | 2.25 (1.05,4.83) | **0.038** |
| **Time spent walking to school** | | | | | | | | |
| None | 1 | | 1 | | 1 | | 1 | |
| 5–30 mins | 1.08 (0.83,1.42) | 0.557 | 0.53 (0.35,0.80) | **0.002** | 0.88 (0.61,1.27) | 0.49 | 3.58 (1.45,8.84) | **0.006** |
| >30 mins | 1.07 (0.75,1.51) | 0.72 | 0.35 (0.17,0.72) | **0.004** | 0.95 (0.59,1.52) | 0.82 | 4.61 (1.75,12.13) | **0.002** |
| **Perform home chores** | | | | | | | | |
| No | 1 | | 1 | | 1 | | 1 | |
| Yes | 1.08 (0.74,1.59) | 0.689 | 0.57 (0.34,0.95) | **0.031** | 1.06 (0.61,1.83) | 0.834 | 1.28 (0.54,3.05) | 0.571 |
| **Household food insecurity** | | | | | | | | |
| Food secure | 1 | | 1 | | 1 | | 1 | |
| Food insecure | 1,21 (0.94,1.56) | 0.13 | 0,51 (0.34,0.75) | **0.001** | 1.05 (0.74,1.48) | 0.8 | 2.10 (1.13,3.91) | **0.019** |
| **Self-rated health** | | | | | | | | |
| Poor/fair | 1 | | 1 | | 1 | | 1 | |
| Good/Excellent | 0.97 (0.72,1.31) | 0.845 | 1.41 (0.74,2.69) | 0.303 | 1.19 (0,73, 1.93) | 0.48 | 0.92 (0.49, 1.72) | 0.791 |
| **Has any chronic illness** | | | | | | | | |
| No | 1 | | 1 | | 1 | | 1 | |
| Yes | 1.19 (0.90,1.58) | 0.219 | 0.86 (0.47,1.59) | 0.643 | 0.89 (0.55,1.44) | 0.633 | 0.87 (0.44,1.75) | 0.705 |

*CPR is crude prevalence ratio, CI- confidence interval and a P-value of <0.05 was considered significant.

*Chronic illnesses including asthma, sickle cell, skin diseases, chronic pain and eye diseases.

**Table 5.** Multivariate analysis of factors associated with anaemia, overweight, stunting and thinness.

| Variables | Anaemia | | Overweight | | Stunting | | Thinness | |
|---|---|---|---|---|---|---|---|---|
| | APR (95% CI) | p-value | APR (95% CI) | p-value | APR (95% CI) | p-value | APR (95% CI) | p-value |
| **Adolescent age** | 0.98 (0.90–1.06) | 0.542 | 1.16 (1.00–1.36) | 0.058 | 1.04 (0.91–1.17) | 0.583 | 1.14 (0.95–1.39) | 0.167 |
| **Household size** | 0.99 (0.94–1.04) | 0.664 | 0.94 (0.84–1.06) | 0.322 | 1.06 (1.01–1.11) | **0.01** | 1.07 (0.97–1.17) | 0.158 |
| **Adolescent Gender** | | | | | | | | |
| Male | 1 | | 1 | | 1 | | 1 | |
| Female | 1.17 (0.90–1.53) | 0.247 | 2.62 (1.56–4.43) | **<0.001** | 0.63 (0.45–0.88) | **0.006** | 0.29 (0.16–0.54) | **<0.001** |
| **Currently attending school** | | | | | | | | |
| No | 1 | | 1 | | 1 | | 1 | |
| Yes | 0.92 (0.55–1.53) | 0.747 | 1.87 (0.75–4.62) | 0.178 | 0.75 (0.39–1.45) | 0.391 | 3.38 (1.0–11.44) | 0.05 |
| **Adolescent school level** | | | | | | | | |
| Primary education | 1 | | 1 | | 1 | | 1 | |
| Secondary/higher | 1.21 (0.86–1.70) | 0.277 | 0.62 (0.32–1.21) | 0.162 | 0.68 (0.40–1.14) | 0.144 | 0.55 (0.25–1.22) | 0.143 |
| **Household Wealth Index** | | | | | | | | |
| Low | 1 | | 1 | | 1 | | 1 | |
| Middle | 1.13 (0.87–1.47) | 0.356 | 1.35 (0.78–2.32) | 0.281 | 0.7 (0.48–1.01) | 0.058 | 0.94 (0.58–1.55) | 0.821 |
| High | 1.3 (0.95–1.76) | 0.101 | 1.14 (0.60–2.18) | 0.692 | 0.74 (0.48–1.13) | 0.167 | 0.39 (0.16–0.96) | **0.04** |
| **Household head education** | | | | | | | | |
| Primary/no formal | 1 | | 1 | | 1 | | 1 | |
| Secondary/higher | 0.82 (0.63–1.07) | 0.138 | 1.05 (0.67–1.65) | 0.823 | 0.95 (0.66–1.37) | 0.775 | 1.14 (0.65–2.01) | 0.637 |
| **Household toilet facility** | | | | | | | | |
| Improved | 1 | | 1 | | 1 | | 1 | |
| Unimproved | 1.51 (1.16–1.95) | **0.002** | 1.31 (0.73–2.35) | 0.367 | 1.13 (0.74–1.71) | 0.58 | 1.1 (0.64–1.90) | 0.725 |
| **Played any sport in the past 7 days** | | | | | | | | |
| No | 1 | | 1 | | 1 | | 1 | |
| Yes | 0.6 (0.11–3.22) | 0.554 | 2.02 (0.77–5.32) | 0.153 | 0.62 (0.20–1.95) | 0.412 | 2.07 (0.26–16.73) | 0.494 |
| **Days spent on physical activity** | | | | | | | | |
| None | 1 | | 1 | | 1 | | 1 | |
| 1-2 | 1.4 (0.26–7.56) | 0.694 | 0.5 (0.18–1.36) | 0.173 | 2.38 (0.75–7.56) | 0.14 | 0.85 (0.11–6.62) | 0.873 |
| 3-5 | 1.37 (0.25–7.45) | 0.714 | 0.33 (0.11–1.00) | 0.05 | 1.53 (0.47–4.96) | 0.477 | 0.76 (0.10–5.87) | 0.789 |
| 6-7 | 1.08 (0.20–5.91) | 0.928 | 0.39 (0.13–1.17) | 0.094 | 1.59 (0.51–4.95) | 0.428 | 0.51 (0.07–3.95) | 0.521 |
| **Time spent walking to school** | | | | | | | | |
| None | 1 | | 1 | | 1 | | 1 | |
| 5-30 mins | 1.15 (0.87–1.51) | 0.325 | 0.75 (0.48–1.19) | 0.228 | 0.82 (0.56–1.20) | 0.31 | 2.66 (1.02–6.94) | **0.046** |
| >30 mins | 1.09 (0.77–1.53) | 0.632 | 0.55 (0.26–1.17) | 0.121 | 0.83 (0.50–1.36) | 0.451 | 3.39 (1.12–10.25) | **0.03** |
| **Perform home chores** | | | | | | | | |
| No | 1 | | 1 | | 1 | | 1 | |
| Yes | 1.1 (0.73–1.66) | 0.641 | 0.73 (0.42–1.26) | 0.258 | 1.21 (0.67–2.20) | 0.524 | 0.76 (0.28–2.07) | 0.591 |
| **Household food insecurity** | | | | | | | | |
| Food secure | 1 | | 1 | | 1 | | 1 | |
| Food insecure | 1.1 (0.84–1.45) | 0.486 | 0.59 (0.38–0.92) | **0.02** | 0.97 (0.67–1.40) | 0.854 | 1.61 (0.85–3.02) | 0.141 |
| **Self-rated health** | | | | | | | | |
| Poor/fair | 1 | | 1 | | 1 | | 1 | |
| Good/Excellent | 1.05 (0.77–1.43) | 0.746 | 1.61 (0.84–3.11) | 0.154 | 1.24 (0.75–2.05) | 0.394 | 0.8 (0.43–1.50) | 0.49 |
| **Has any chronic illness** | | | | | | | | |
| No | 1 | | 1 | | 1 | | 1 | |
| Yes | 1.11 (0.83–1.48) | 0.481 | 0.91 (0.48–1.72) | 0.776 | 0.77 (0.45–1.30) | 0.329 | 1 (0.48–2.09) | 0.992 |

*(Continued)*

 

**Table 5.** (Continued)

*Adjusted for all predictor variables.

APR – Adjusted prevalence ratio, CI- confidence interval and a P-value of <0.05 was considered significant.

*Chronic illnesses including asthma, sickle cell, skin diseases, chronic pain and eye diseases.

Addressing household food insecurity in urban Tanzanian settings could involve innovative social protection mechanisms such as conditional cash transfers, which have proven effective in reducing household vulnerability to poverty in Tanzania [33]. In addition, exploring targeted food provision programs for adolescents, particularly those out of school, may ensure adequate nutritional intake.

Male adolescents and those from larger household sizes had a higher risk of stunting. Stunting not only impairs growth and development but also has long-term consequences on health and human capital, limiting children's potential and perpetuating cycles of poverty [1,8]. This observation underscores the importance of addressing socioeconomic inequities, particularly in larger urban households, which are more prone to the effects of poverty [34].

Anaemia was also highly prevalent among adolescent girls, with one in five experiencing moderate or severe anaemia. This finding is consistent with existing literature and has been associated with increased iron demands during menarche [8]. To improve iron levels, intermittent iron supplementation is a cost-effective strategy for preventing and reducing anaemia in this population [35]. Additionally, incorporating iron-fortified foods, such as cereals and condiments, has effectively reduced anaemia in different settings [36,37]. In Tanzania, mandatory fortification policies for cereals such as wheat and maize are in place. However, their effectiveness is limited by the high prevalence of homegrown cereal consumption. Policymakers must develop innovative strategies to extend the benefits of fortification to households relying on home-produced cereals [38].

The strong association between unimproved toilet facilities and anaemia in this study aligns with evidence linking poor sanitation to an increased risk of helminth infections, which contribute to iron deficiency and anaemia [39]. This underscores the need for comprehensive sanitation and hygiene interventions. Helminth control programs, such as regular deworming campaigns in schools, should be prioritised as a cost-effective strategy to reduce the burden of anaemia among adolescents [40,41]. Furthermore, investments in improved sanitation infrastructure, particularly in urban low-income settings, could have far-reaching health benefits.

However, gender is not solely a biological factor but a structural determinant of malnutrition, shaping girls' access to nutritious food, health care, and clean sanitation through restrictive gender norms and systemic inequality [42]. As such, policies must go beyond biological considerations and target these structural barriers. In urban Tanzanian settings, gender-sensitive strategies should be integrated into anaemia prevention programs to ensure adolescent girls have equitable access to iron-rich foods, health education, and sanitation services, while also addressing the underlying gender norms that limit access to these essential resources.

Policymakers should adopt intersectional and integrated approaches to reduce poverty and improve adolescent nutrition. Combined strategies such as cash transfers and public work schemes could reduce the number of households with unimproved toilet facilities, food insecurity, and low income, ultimately improving adolescents' nutritional outcomes in Dar es Salaam and similar settings [33].

As a cross-sectional study, this research contributes to filling data gaps on adolescent malnutrition in urban LMICs and highlighting gendered vulnerabilities and their implications for targeted interventions. This research is particularly relevant in Dar es Salaam, where rapid urbanisation, social inequality, and dietary transitions converge to heighten adolescents' vulnerability to malnutrition [34]. By generating data that highlights gendered nutritional disparities, the study supports national and global efforts to reduce adolescent malnutrition and address the nutritional needs of adolescent girls as outlined in the WHO Global Nutrition Targets and Sustainable Development Goals [43].

This study used a household survey to ensure a representative sample of in-school and out-of-school adolescents. Anthropometric measurements were taken twice for each participant to minimise the possible measurement error. Despite its strengths, the present study has its limitations. Firstly, our study only focused on the co-existing forms of malnutrition in the community and not in an individual due to a limited sample size. Secondly, the findings are only generalizable to the urban adolescent population. Thirdly, the study could not account for the duration or intensity of physical activity, which could explain the lack of association with the outcomes in the multivariate analysis. Also, this is a cross-sectional study; hence, we cannot draw any conclusions regarding causation. Future research should explore longitudinal designs to better understand the causative factors of adolescent malnutrition and assess the long-term impact of interventions. Information bias is also possible due to socially desirable answers to lifestyle and behavioural questions. To minimise this bias, researchers were trained to create a good rapport. Lastly, the study did not collect information on menstrual cycle phase or pubertal development, which may have introduced some measurement variability in haemoglobin and anthropometric outcomes. Future studies may benefit from incorporating these variables to better account for physiological changes during adolescence.

## Conclusion

In conclusion, adolescent malnutrition in Dar es Salaam remains high, associated with food insecurity, poor sanitation, and gender inequities. Addressing these challenges requires integrated, equity-focused policies that strengthen nutrition, improve WASH services, and promote gender equality. Such actions, aligned with SDGs 2, 3, 5, and 6, are essential to close health gaps and ensure all adolescents can achieve their full potential.

## Supporting information

**S1 File. Study questionnaire.**
(PDF)

**S1 Checklist. Inclusivity in global research checklist.**
(DOCX)

## Acknowledgments

We sincerely appreciate the adolescents and their guardians who graciously agreed to participate in our study. We also extend our gratitude to our dedicated data collectors, whose hard work and commitment were instrumental in making this research possible. Additionally, we are thankful to our colleagues at the University of Bergen, Norway and the Muhimbili University of Health and Allied Sciences, Tanzania for their valuable cooperation, with special recognition to Dr. Jackline Ngowi for her guidance and support in obtaining ethical clearances in Tanzania. We also deeply appreciate Stanslaus Henry for his generous support during data collection and analysis. Lastly, we acknowledge the contributions of our colleagues from the National Bureau of Statistics (Tanzania) and the Tanzania Food and Nutrition Centre for their assistance with survey logistics.

## Author contributions

**Conceptualization:** Christina Kimaryo, Josephine Efraim, Nancy Njenge, Bruno Sunguya.

**Data curation:** Christina Kimaryo, Josephine Efraim, Nancy Njenge.

**Formal analysis:** Christina Kimaryo.

**Funding acquisition:** Lyn Haskins, Anne Hatløy, Christiane Horwood, Bruno Sunguya.

**Investigation:** Christina Kimaryo.

**Methodology:** Christina Kimaryo, Josephine Efraim.

**Project administration:** Anne Hatløy.

**Resources:** Lyn Haskins, Anne Hatløy, Bruno Sunguya.

**Software:** Christina Kimaryo.

**Supervision:** Christina Kimaryo, Lyn Haskins, Anne Hatløy, Christiane Horwood.

**Validation:** Christina Kimaryo.

**Writing – original draft:** Christina Kimaryo, Josephine Efraim, Nancy Njenge, Bruno Sunguya.

**Writing – review & editing:** Christina Kimaryo, Lyn Haskins, Anne Hatløy, Christiane Horwood.

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
