## [Decision Letter · Decision Letter 0]

19 Jun 2025

Thank you for submitting your manuscript to PLOS ONE. After careful consideration, we feel that it has merit but does not fully meet PLOS ONE’s publication criteria as it currently stands. Therefore, we invite you to submit a revised version of the manuscript that addresses the points raised during the review process.

We look forward to receiving your revised manuscript.

Kind regards,

Bilal Ahmad Rahimi, M.D., D.T.M.&H., M.C.T.P., Ph.D

Academic Editor

PLOS ONE

Journal Requirements:

[This study was conducted as part of EPRENUT, a NORPART project funded by HK Dir Norway (NORPART-2021/10434). The funders had no role in study design, data collection and analysis, decision to publish, or preparation of the manuscript.].

4. In the online submission form, you indicated that [Data cannot be shared publicly because of the data-sharing agreement signed when obtaining ethical permission from ethical boards in Tanzania. Data are available from the University of Bergen's safe server and can be obtained under special request to the ethical boards.].

5. Please amend the manuscript submission data (via Edit Submission) to include author Christina Kimaryo.

6. Please amend your authorship list in your manuscript file to include author Christina Judathadei Kimaryo.

Reviewers' comments:

Reviewer's Responses to Questions

**Comments to the Author**

1. Is the manuscript technically sound, and do the data support the conclusions?

Reviewer #1: Yes

Reviewer #2: Partly

2. Has the statistical analysis been performed appropriately and rigorously?

Reviewer #1: Yes

Reviewer #2: No

3. Have the authors made all data underlying the findings in their manuscript fully available?

Reviewer #1: Yes

Reviewer #2: No

4. Is the manuscript presented in an intelligible fashion and written in standard English?

Reviewer #1: Yes

Reviewer #2: Yes

Reviewer #1: well written. need to emphasise the limitations of a crosssectional design for casual research more clearly . thestudy otherwise is well written and adresses an important area of health research which needs to be highlighted

Reviewer #2: The relevance of the topic in adolescent health fills essential gaps in missing information in LMICs. I’ll encourage the authors to reconsider some points in their document to improve it.

Thank you!

1. Keywords and Framing

Recommendation: Include keywords such as gender, low- and middle-income countries (LMICs), and vulnerable populations to enhance the article’s discoverability and accurately reflect its analytical scope.

Critical Note: The manuscript should adopt an intersectional framework from the outset, positioning gender, socioeconomic deprivation, and adolescent malnutrition as interlinked determinants. This framing should guide the research question, rationale, and interpretation throughout the paper.

2. Introduction

Lines 43–44: Clarify the distinction between physical growth (e.g., linear height, weight gain) and development (e.g., cognitive, emotional, and psychosocial maturation).

Recommendation: Introduce the triple burden of malnutrition—undernutrition, micronutrient deficiencies, and overweight/obesity—early in the text, particularly as it affects adolescents in resource-constrained settings.

Gap: Explicitly address how gender inequities and structural policy shortcomings contribute to adolescent malnutrition in urban, deprived environments. This is essential to contextualize the study's added value.

3. Methodology

a. Study Design

Line 85: Clearly describe the cross-sectional analytical design used. Include references that support the development and validation of the data collection instrument, and describe the steps taken during pre-testing or piloting.

Provide details on enumerator training, field protocols, and quality assurance measures to support claims of internal and external validity.

b. Sampling and Participants

Justify the age range selected, particularly in relation to pregnant and postpartum adolescents, whose nutritional and physiological profiles may differ from non-pregnant peers.

Clarify the sampling frame, cluster selection criteria, and how this sample compares to the general adolescent population in Dar es Salaam.

Avoid fragmentation in the description of inclusion/exclusion criteria and participant characteristics. Reorganize these sections to improve flow and comprehension.

c. Data Collection

Lines 84–89: Clarify distinctions between household-level (family) and individual-level (adolescent) sociodemographic data. If caregiver data contributed to malnutrition and multimorbidity analyses, clearly state this and justify its inclusion.

Articulate the conceptual framework for analyzing the interrelationship between malnutrition indicators (e.g., stunting, thinness, anemia), particularly when these co-occur.

Describe specific controls used to reduce measurement bias from physiological and developmental variability—for example: Adjustment for menstrual cycle timing when measuring anemia in girls;

Use of Tanner staging to contextualize thinness during puberty.

Measurement Tools:

Describe tools used to assess physical activity, referencing validated frameworks (e.g., "Assessment of physical activity among adolescents: a guide to the literature – PMC") and clarify whether this variable was used in the multivariate models.

Detail how wealth index, food insecurity, chronic diseases, disabilities, and mental health were operationalized. Consider disaggregating mental health outcomes if their associations with malnutrition differ.

Define and standardize all covariates used in model adjustments.

For anemia assessment, report on measurement accuracy, precision, tool sensitivity to temperature, and whether altitude adjustments were applied.

Line 154: Correct formatting of bold references.

Justify the decision to collapse food insecurity categories—was this based on statistical power or conceptual reasoning?

Analytical Approach:

Consider a multilevel modeling approach to appropriately account for the hierarchical nature of data (individuals nested within households).

4. Results Interpretation

Recommendation: Reduce the number of tables and concentrate on presenting the most relevant findings. Results should be interpreted through an intersectional lens that considers how gender, socioeconomic conditions, and nutritional outcomes are interrelated.

Provide contextual data: For comparative purposes, cite national or regional statistics on adolescent malnutrition in Dar es Salaam or Tanzania in the introduction or results section.

a. Nutritional Outcomes

Discuss how pubertal development stages (e.g., Tanner stages) influence thinness or overweight prevalence.

Clearly define co-occurrence or coexistence of malnutrition outcomes, including the time frame used to assess simultaneity.

Describe how overlapping forms of malnutrition were analyzed within a physio-pathological framework, addressing potential shared or opposing mechanisms.

b. Gender Analysis

Interpret higher prevalence of anemia or thinness in girls through a gendered lens, considering cultural norms, caregiving roles, parasitic exposure, and dietary practices.

Lines 257, 264–277: Consolidate repetitive content regarding causes of anemia to streamline the narrative and improve clarity.

c. Table Improvements

Table 2: Add assumptions and interpretation guidance for Chi-square tests.

Table 6: List the specific chronic illnesses and disabilities reported by participants.

Table 7: Clarify how physical activity levels were cross-tabulated with thinness categories.

Across all tables: Prioritize clarity and simplicity. Explain how key covariates were operationalized.

Define how coexisting forms of malnutrition were captured at both individual and household levels.

5. Discussion and Contribution

Structure: The discussion lacks coherence and leans heavily on descriptive repetition. Shift toward interpretive analysis that links results to broader public health and policy contexts.

Avoid speculative explanations for health outcomes (e.g., anemia) without data support.

Clarify the study’s relevance: Why is this research important in Dar es Salaam, and how does it contribute to global efforts to reduce adolescent malnutrition?

Emphasize the study’s contribution to addressing gender and nutrition equity, particularly as a cross-sectional analysis.

Treat gender as a structural determinant of malnutrition, not simply a demographic characteristic—especially given its prominence in the title.

Summarize contributions in terms of: Filling data gaps on adolescent malnutrition in urban LMICs,

Highlighting gendered vulnerabilities and their implications for targeted interventions.

6. Final Recommendations

Anchor results and discussion within a conceptual framework that integrates gender, socioeconomic status, physiological development, and environmental exposures.

Consider incorporating: Behavioral models related to diet and physical activity; Intersectionality frameworks to inform policy recommendations; Multilevel statistical models to distinguish household and individual effects.

Conclusion: Should synthesize key findings, their relevance to closing equity gaps, and outline practical implications for nutrition and health policy, especially in alignment with Sustainable Development Goals (SDGs).

**Do you want your identity to be public for this peer review?** For information about this choice, including consent withdrawal, please see our Privacy Policy

Reviewer #1: No

Reviewer #2: No

---

## [Author Response · Author response to Decision Letter 1]

22 Aug 2025

Thank you for the valuable feedback provided by the reviewers, which has helped improve the manuscript's quality. Below are detailed responses to each comment, and all revisions have been marked within the text.

1. Keywords and Framing

1a. Recommendation: Include keywords such as gender, low- and middle-income countries (LMICs), and vulnerable populations to enhance the article’s discoverability and accurately reflect its analytical scope.

Author's response: We agree that including these keywords will improve the manuscript's visibility and better represent its focus. Accordingly, we have added “gender,” “low- and middle-income countries (LMICs),” and “vulnerable populations” to the list of keywords.

1b. Critical Note: The manuscript should adopt an intersectional framework from the outset, positioning gender, socioeconomic deprivation, and adolescent malnutrition as interlinked determinants. This framing should guide the research question, rationale, and interpretation throughout the paper.

Author´s response: Thank you for the comment. We have restructured our manuscript, accordingly, as shown in the 3rd paragraph of the introduction section.

2. Introduction

2a. Lines 43–44: Clarify the distinction between physical growth (e.g., linear height, weight gain) and development (e.g., cognitive, emotional, and psychosocial maturation).

Author's response: Thank you for your comment. We have added the distinction in the first paragraph of the introduction lines 44-48

2b. Recommendation: Introduce the triple burden of malnutrition—undernutrition, micronutrient deficiencies, and overweight/obesity—early in the text, particularly as it affects adolescents in resource-constrained settings.

Author's response: We agree that it is important to introduce this concept earlier in the text. We have added the definition of the triple burden of malnutrition in the second paragraph of the introduction from line 50.

2c. Gap: Explicitly address how gender inequities and structural policy shortcomings contribute to adolescent malnutrition in urban, deprived environments. This is essential to contextualise the study's added value.

Author's response: Thank you for pointing this out. We have added this to the manuscript in lines 59-62.

3. Methodology

3a. Study Design

3a1. Line 85: Clearly describe the cross-sectional analytical design used. Include references that support the development and validation of the data collection instrument, and describe the steps taken during pre-testing or piloting.

Author's response: We have added this information to our revised manuscript in the data collection section In lines 134-136 and 140-142.

3a2. Provide details on enumerator training, field protocols, and quality assurance measures to support claims of internal and external validity.

Author's response: We agree that this is important information; we have added it to the manuscript in lines 131-134 and 156-159.

3b. Sampling and Participants

b1. Justify the age range selected, particularly in relation to pregnant and postpartum adolescents, whose nutritional and physiological profiles may differ from non-pregnant peers.

Author's response: Thank you for bringing this important point to our attention. The age range of 12–19 years was selected as it spans the full spectrum of adolescence, from early to late stages, capturing the wide variability in physical development that occurs during this period. We used age-specific measures to assess nutritional status (e.g., weight-for-height Z-scores, age-specific haemoglobin cut-offs), ensuring that developmental differences were appropriately accounted for in the analysis. We excluded pregnant and postpartum adolescents from the study to minimise potential bias, given that their nutritional and physiological needs may differ significantly from those of their non-pregnant peers. This is written in the study population section in lines 106-109.

3b2. Clarify the sampling frame, cluster selection criteria, and how this sample compares to the general adolescent population in Dar es Salaam.

Author's response: Thank you for bringing this to our attention. We have added these details to our revised manuscript, lines 116-120.

3b3. Avoid fragmentation in the description of inclusion/exclusion criteria and participant characteristics. Reorganize these sections to improve flow and comprehension.

Author's response: Thank you for the comment. The method section has been edited to improve the flow.

3c. Data Collection

3c1. Lines 84–89: Clarify distinctions between household-level (family) and individual-level (adolescent) sociodemographic data. If caregiver data contributed to malnutrition and multimorbidity analyses, clearly state this and justify its inclusion.

Author's response: Thank you for this valuable comment. We would like to clarify that caregivers only provided information on household-level characteristics, such as household size, parental education, and household assets. They did not respond to questions related to adolescent-level sociodemographic data or health outcomes. All individual-level data, including those used in the analyses of malnutrition and multimorbidity, were obtained directly from the adolescents themselves. We have revised the Methods section to clarify this distinction in lines 141-144.

3c2. Articulate the conceptual framework for analyzing the interrelationship between malnutrition indicators (e.g., stunting, thinness, anemia), particularly when these co-occur.

Author.s response: Thank you for your thoughtful comment. Our analysis was guided by the UNICEF conceptual framework for factors affecting adolescent nutrition, which informed our selection and categorisation of nutritional indicators and explanatory variables. However, we did not conduct an analytical investigation into the interrelationships or shared risk factors among stunting, thinness, and anaemia. Instead, we focused on estimating the burden of each indicator individually, as well as reporting the co-occurrence of two or more forms of malnutrition as a descriptive measure of nutritional multimorbidity. We have added the information on the manuscript lines 160-161

3c3. Describe specific controls used to reduce measurement bias from physiological and developmental variability—for example: Adjustment for menstrual cycle timing when measuring anemia in girls;

Use of Tanner staging to contextualize thinness during puberty.

Author´s response: Thank you for this thoughtful comment. We acknowledge that physiological and developmental factors such as menstrual cycle timing and pubertal stage can influence haemoglobin levels and anthropometric indicators. However, our household survey did not collect data on Tanner staging and menstrual cycle timing, primarily due to the sensitive nature of these topics and the need to minimise respondent burden in a community-based setting, respectively. We have now acknowledged this in the Limitations section lines 379-382 and have recommended that future studies consider including such measures to improve the accuracy of nutritional assessments.

3d.Measurement Tools:

3d1. Describe tools used to assess physical activity, referencing validated frameworks (e.g., "Assessment of physical activity among adolescents: a guide to the literature – PMC") and clarify whether this variable was used in the multivariate models.

Author's response: Thank you for this helpful comment. We have clarified in the Methods section that the physical activity questions were adapted from the validated questionnaire used in the International Study of Childhood Obesity, Lifestyle and Environment (ISCOLE) (lines 172-173). Additionally, we have written in the manuscript that all variables were used in multivariate models (line 189-190).

3d2. Detail how wealth index, food insecurity, chronic diseases, disabilities, and mental health were operationalized. Consider disaggregating mental health outcomes if their associations with malnutrition differ.

Author's response: Thank you for your valuable feedback. The variables for chronic diseases and self-rated health were analysed in the same form as they were collected in the questionnaire, without recoding or transformation. Specifically, participants responded to structured questions regarding the presence or absence of chronic diseases and their self-rated health status, and these responses were directly used in the analysis and tabulation. Food insecurity was explained in lines 161-166, and wealth index in lines 194-202.

Regarding mental health, disaggregation was not feasible since only 4 participants reported being diagnosed.

3d3. Define and standardise all covariates used in model adjustments.

Author's response: Thank you for the comment, we have added a table (table 1) with these details under the analysis section.

3d4. For anemia assessment, report on measurement accuracy, precision, tool sensitivity to temperature, and whether altitude adjustments were applied.

Author response: Thank you for your helpful comment. We have revised the Methods section to include additional details on haemoglobin measurement. We used two types of Hemocue devices: Hemocue Hb 201+ and Hemocue Hb 801. Since data collection took place outdoors in Dar es Salaam, we took care to conduct measurements in shaded areas and to limit device exposure to direct sunlight and extreme heat, in accordance with manufacturer guidelines. No altitude adjustments were applied, as Dar es Salaam is located at sea level. These details have been added to the revised Methods section lines 150-156.

3d5. Line 154: Correct formatting of bold references.

Author.s response: Thank you for the comment, we have corrected it.

3d6. Justify the decision to collapse food insecurity categories—was this based on statistical power or conceptual reasoning?

Author's response: The original food insecurity variable had four categories: food secure, mildly food insecure, moderately food insecure, and severely food insecure. For analysis, we collapsed these into two categories (food secure vs. food insecure) due to small sample sizes in some subgroups (eg. the mildly food insecure subgroup), which limited statistical power for detecting meaningful differences across all four levels. We have added the justification to the manuscript lines 168-170.

3e.Analytical Approach:

3e1. Consider a multilevel modeling approach to appropriately account for the hierarchical nature of data (individuals nested within households).

Author's response: Thank you for the recommendation. However, in our study, we did not use a nested data structure. Instead, we merged household-level data with adolescent-level data and included only one adolescent per household in the final analysis. As a result, there is no clustering of individuals within households, and the data are treated at a single level. Therefore, a multilevel modelling approach was not necessary in this case.

4. Results Interpretation

4a Recommendation: Reduce the number of tables and concentrate on presenting the most relevant findings. Results should be interpreted through an intersectional lens that considers how gender, socioeconomic conditions, and nutritional outcomes are interrelated.

Author's response: Thank you for the recommendation. We have merged some tables to reduce the number of tables and revised the results interpretation as suggested.

4b Provide contextual data: For comparative purposes, cite national or regional statistics on adolescent malnutrition in Dar es Salaam or Tanzania in the introduction or results section.

Author's response: We agree that contextual data are important; we provided data on Dar es Salaam in lines 77-80 in the introduction section of the manuscript.

4c. Nutritional Outcomes

4c1. Discuss how pubertal development stages (e.g., Tanner stages) influence thinness or overweight prevalence.

Author's response: Thank you for this valuable comment. Our study did not assess pubertal development stages (e.g., Tanner stages), and therefore we were unable to directly examine their influence on thinness or overweight prevalence in our discussion. We agree that pubertal development may play an important role in adolescent nutritional outcomes, as physiological changes during puberty can affect body composition, growth velocity, and energy requirements. We have now included this point as a recommendation (lines 3798-382) for future research, suggesting that studies incorporate assessments of pubertal stage to better account for developmental differences when examining malnutrition in adolescents.

4c2. Clearly define co-occurrence or coexistence of malnutrition outcomes, including the time frame used to assess simultaneity.

Author's response: Thank you for your comment. In our analysis, co-occurrence (or coexistence) of malnutrition outcomes refers to the presence of two or more forms of malnutrition (e.g., stunting and overweight) in the same individual at the same time. All anthropometric measurements and blood tests used to determine these outcomes were collected during the same time in every individual, and thus reflect a single time point, allowing us to assess simultaneity directly. We have now clarified this in lines 239-240.

4c3 Describe how overlapping forms of malnutrition were analyzed within a physiopathological framework, addressing potential shared or opposing mechanisms.

Authors comment: Thank you for your insightful comment. In our study, due to a smaller sample size, we focused on analysing the individual burden of each form of malnutrition (e.g., anemia, stunting, thinness) separately. We did not examine overlapping or co-existing forms of malnutrition within a physiopathological framework, nor did we analyse shared or opposing mechanisms between them. We acknowledge this as a potential area for future research and have clarified this limitation in the revised manuscript lines (370-371).

4d. Gender Analysis

4d1 Interpret higher prevalence of anemia or thinness in girls through a gendered lens, considering cultural norms, caregiving roles, parasitic exposure, and dietary practices.

Author.s response: Thank you for the comment, we have incorporated these points to our revised manuscript.

4d2 Lines 257, 264–277: Consolidate repetitive content regarding causes of anaemia to streamline the narrative and improve clarity.

Author's response: Thank you for bringing this to our attention. We have consolidated the repetitive anameia content to improve narrative and clarity as suggested.

4e. Table Improvements

4e1 Table 2: Add assumptions and interpretation guidance for Chi-square tests.

Author's response: We have added the assumption to Table 3 since Table 2 contains only descriptive statistics.

4e2 Table 6: List the specific chronic illnesses and disabilities reported by participants.

Authors response: Thank you for the comment, we have added these details

4e3 Table 7: Clarify how physical activity levels were cross-tabulated with thinness categories.

Authors response: Thank you for the comment, physical activity categories were analysed as they are shown in the table, no further modification to the variables were made.

4e4. Across all tables:

Prioritize clarity and simplicity.

Author´s response: Done

Explain how key covariates were operationalised.

Author´s response: Thank you for the comment. We have explained in the method section lines 177-181

4e5. Define how coexisting forms of malnutrition were captured at both individual and household levels.

Author´s response: Thank you for bringing this to our attention. We have added the information in lines 239-240

5. Discussion and Contribution

Structure: The discussion lacks coherence and leans heavily on descriptive repetition. Shift toward interpretive analysis

---

## [Decision Letter · Decision Letter 1]

19 Sep 2025

Dear Dr. Kimaryo,

Thank you for submitting your manuscript to PLOS ONE. After careful consideration, we feel that it has merit but does not fully meet PLOS ONE’s publication criteria as it currently stands. Therefore, we invite you to submit a revised version of the manuscript that addresses the points raised during the review process.

We look forward to receiving your revised manuscript.

Kind regards,

Bilal Ahmad Rahimi, M.D., D.T.M.&H., M.C.T.P., Ph.D

Academic Editor

PLOS ONE

Journal Requirements:

Reviewers' comments:

Reviewer's Responses to Questions

**Comments to the Author**

Reviewer #3: (No Response)

2. Is the manuscript technically sound, and do the data support the conclusions?

Reviewer #3: Partly

3. Has the statistical analysis been performed appropriately and rigorously?

Reviewer #3: No

4. Have the authors made all data underlying the findings in their manuscript fully available?

Reviewer #3: Yes

5. Is the manuscript presented in an intelligible fashion and written in standard English?

Reviewer #3: Yes

Reviewer #3: GENERAL COMMENTS/QUERIES/RECOMMENDATIONS:

1. The manuscript contents pinpoint an important global health concern.

2. The study design and even the quantitative nature of findings provides real-time data for comparison if done appropriately and correctly.

3. However, there are a number of lexicographic, syntactic as well as logical flaws in almost every section of this manuscript

e.g.: Dar es Salaam city has been judiciously divided into five ‘municipalities’ and not councils as suggested by the authors!

Otherwise, Dar es Salaam city (currently a metropolis proper!) has been evidently reported to have > 5.3 million residents in the most recent Population and Housing Census data conducted in 2022 and reported in 2023. In fact, very recent CIA factsheet estimates approximates the findings to be > 7 million (by July 2024) and hence, it is questionable where did authors get the 'magic number' of 1.6 million?

Specific Queries/Comments/Recommendations:

1. The fact that there has been no definition of the term ‘adolescence’ in the entire manuscript, with authors defining the age group (of 13-19 years) in the study settings sub-section (pp. 9 line 97), can I believe this was the definition of ‘adolescence’ in this study? If so, how different is this definition (based on the specified age group) different from that of ‘teenage’? If not, what was the defining term of the word ‘adolescence’ by the authors??

Recommendations: authors are advised to define what they meant by the term ‘adolescence’ somewhere in the introduction section, in order to avoid unnecessary confusion associated with the blind usage of the term in general literature.

Evidence: adolescence has been defined differently by different scholars. For those in social sciences/demographers, it normally entails a period between 13-17 years, those in biogerontology accepting age threshold of 10-24 years, e.t.c.

2.It was not clear what was the primary sampling unit in this study between ‘households’ or ‘adolescents’? Authors should elaborate it clearly!

Evidence: whereas there is evidence of adolescents aged 12-19 years (pp. 9, line 104) as the study population, 12 households in each randomly chosen cluster (pp. 10, line 115), it became apparent that in actual fact, what any given reader would consider to be a total sample size of 507 to be adolescents studied, in fact it was the total number of households!

(refer pp. 16, line 226)

3.Why did authors opt for a ‘modified Poisson regression model’ amidst evidence that background studies in the introduction section already suggested the burden of malnutrition (prevalence of anaemia, stunting to be 34%, 16% respectively) among adolescents neither to be ‘count data’ nor ‘rare events’?

Reason: by default, before investigators/authors decide to fit a Poisson regression model if they are dealing with ‘count data’ (which I do not think any of the assessed variables were count data anyways!) as well as justifying assumptions of ‘Poisson regression model’ have been made.

4. Why did authors opt to analyse association between ‘overweight’, ‘thinness’, ‘stunting’ and ‘anaemia’ using Pearson Chi-square test?

Reason: Pearson chi-square test is valid in assessing association among ‘continuous data’ only. None of the stated variables were ‘continuous data’ in the reported manuscript!

5. Why didn’t authors also include a test for interation(s) (effect modification) in their final analysis if there seems to be evidence of a possible statistically significant interaction between the factors ‘overweight’ and ‘thinness’?

Recommendations: Authors are advised to consult ‘a chartered statistician’ or ‘qualified biostatistician’ for assistance on the matter!

**Do you want your identity to be public for this peer review?** For information about this choice, including consent withdrawal, please see our Privacy Policy

Reviewer #3: **Yes: ** Kelvin Melkizedeck Leshabari

---

## [Author Response · Author response to Decision Letter 2]

3 Nov 2025

We thank the reviewers for their constructive feedback. In addition to this response, we have provided detailed responses in the rebuttal letter.

1. REVIEWER COMMENT: Dar es Salaam city has been judiciously divided into five ‘municipalities’ and not councils as suggested by the authors!

Otherwise, Dar es Salaam city (currently a metropolis proper!) has been evidently reported to have > 5.3 million residents in the most recent Population and Housing Census data conducted in 2022 and reported in 2023. In fact, very recent CIA factsheet estimates approximates the findings to be > 7 million (by July 2024) and hence, it is questionable where did authors get the magic number of 1.6 million?

AUTHORS' RESPONSE: We thank the reviewer for this insightful comment. We would like to clarify that in this study, the term councils refers to the five administrative units within Dar es Salaam Region Dar es Salaam City Council (formerly Ilala Municipal Council), Kinondoni, Temeke, Ubungo, and Kigamboni—which are officially recognised as municipal councils under Tanzania’s local government structure.

Regarding the population figure, the number of 1.6 million refers specifically to the Dar es Salaam City Council (Ilala Municipality), where the study was conducted, rather than the entire Dar es Salaam Region. According to the 2022 Population and Housing Census (National Bureau of Statistics, reported in 2024), the Dar es Salaam Region indeed has a population exceeding 5.3 million. However, the population of the Dar es Salaam City Council (Ilala Municipality) is reported as approximately 1.6 million in the same source (NBS, 2024; link to the census data).

2. REVIEWER COMMENT: the fact that there has been no definition of the term ‘adolescence’ in the entire manuscript, with authors defining the age group (of 13-19 years) in the study settings sub-section (pp. 9 line 97), can I believe this was the definition of ‘adolescence’ in this study? If so, how different is this definition (based on the specified age group) different from that of ‘teenage’? If not, what was the defining term of the word ‘adolescence’ by the authors??

AUTHORS' RESPONSE: We thank the reviewer for this thoughtful comment. In our study, we defined adolescence as individuals aged 12–19 years (as shown in the study population section). While the World Health Organisation defines adolescence as the period between 10 and 19 years, we excluded those aged 10–11 years because studies show that adolescents from the age of 12 may have the capacity to be decision-making competent, hence likely to make an informed participation compared to younger ones. We have now clarified this and defined adolescence explicitly in the introduction lines 49-52 to avoid ambiguity, and have added a reference to back up our age selection.

3. REVIEWER COMMENT: .It was not clear what was the primary sampling unit in this study between ‘households’ or ‘adolescents’? Authors should elaborate it clearly!

Evidence: whereas there is evidence of adolescents aged 12-19 years (pp. 9, line 104) as the study population, 12 households in each randomly chosen cluster (pp. 10, line 115), it became apparent that in actual fact, what any given reader would consider to be a total sample size of 507 to be adolescents studied, in fact it was the total number of households!

(refer pp. 16, line 226)

AUTHORS' RESPONSE: We thank the reviewer for this important comment. To clarify, our study employed a multi-stage cluster sampling design, in which clusters were the first-stage primary sampling units (PSUs). Within each selected cluster, 12 households were randomly selected, and from each household, one eligible adolescent aged 12–19 years was randomly chosen to participate, resulting in a total sample of 507 adolescents. We have reviewed the methods section to make it clear (lines 120-124).

4. REVIEWER COMMENT: Why did authors opt for a ‘modified Poisson regression model’ amidst evidence that background studies in the introduction section already suggested the burden of malnutrition (prevalence of anaemia, stunting to be 34%, 16% respectively) among adolescents neither to be ‘count data’ nor ‘rare events’?

Reason: by default, before investigators/authors decide to fit a Poisson regression model if they are dealing with ‘count data’ (which I do not think any of the assessed variables were count data anyways!) as well as justifying assumptions of ‘Poisson regression model’ have been made.

AUTHORS' RESPONSE: We thank the reviewer for this insightful comment and the opportunity to clarify our choice of statistical model. We acknowledge that a standard Poisson regression is typically used for count data. However, in this study, we employed a modified Poisson regression model with robust error variance, which is appropriate for binary outcome variables in cross-sectional studies when the outcome is not rare.

The modified Poisson approach was used to directly estimate prevalence ratios (PRs) rather than odds ratios (ORs), which can overestimate associations when outcomes are common (prevalence >10%). Although the study outcomes (e.g., anaemia, stunting, thinness, overweight/obesity) were binary rather than traditional count data, they can be conceptualised as counts within a contingency table (e.g., number of adolescents with or without anaemia by gender or food insecurity status). In this context, Poisson log-linear models are theoretically appropriate for modelling such data, which further supports the validity of using a modified Poisson regression with robust variance in this study.

5. REVIEWER COMMENT: Why did authors opt to analyse association between ‘overweight’, ‘thinness’, ‘stunting’ and ‘anaemia’ using Pearson Chi-square test?

Reason: Pearson chi-square test is valid in assessing association among ‘continuous data’ only. None of the stated variables were ‘continuous data’ in the reported manuscript!

AUTHORS' RESPONSE: We thank the reviewer for their comment. We would like to clarify that the Pearson Chi-square test is appropriate for assessing associations between categorical variables, such as overweight, thinness, stunting, and anaemia in our study. The reviewer may have confused the Pearson Chi-square test with the Pearson product-moment correlation, which is used for continuous variables. In contrast, the Chi-square test compares observed versus expected frequencies in a contingency table and is standard practice for categorical data.

6. REVIEWER COMMENT: Why didn’t authors also include a test for interation(s) (effect modification) in their final analysis if there seems to be evidence of a possible statistically significant interaction between the factors ‘overweight’ and ‘thinness’?

Recommendations: Authors are advised to consult ‘a chartered statistician’ or ‘qualified biostatistician’ for assistance on the matter!

AUTHORS' RESPONSE: We thank the reviewer for this insightful comment regarding potential interaction (effect modification) between overweight and thinness. In our analysis, overweight and thinness were treated as mutually exclusive categorical outcomes, meaning that a child could not simultaneously be both overweight and thin. Therefore, statistical interaction between these two variables is not applicable in this context. However, we agree that assessing interactions is important in general.

---

## [Editor Report · Decision Letter 2]

4 Nov 2025

The Triple Burden of Malnutrition Among Adolescents in Dar es Salaam, Tanzania: The Role of Gender, Household Environment, and Food Insecurity

PONE-D-25-13116R2

Dear Dr. Christina,

We’re pleased to inform you that your manuscript has been judged scientifically suitable for publication and will be formally accepted for publication once it meets all outstanding technical requirements.

Kind regards,

Bilal Ahmad Rahimi, M.D., D.T.M.&H., M.C.T.P., Ph.D

Academic Editor

PLOS ONE

Additional Editor Comments (optional):

Hello,

Now the Manuscript looks much better and is ready for acceptance.

Best wishes

Bilal
---

## [Editor Report · Acceptance letter]

PONE-D-25-13116R2

PLOS ONE

Dear Dr. Kimaryo,

I'm pleased to inform you that your manuscript has been deemed suitable for publication in PLOS ONE. Congratulations! Your manuscript is now being handed over to our production team.

Kind regards,

on behalf of

Professor Bilal Ahmad Rahimi

Academic Editor

PLOS ONE